# Digested Civet Coffee Beans (Kopi Luwak)—An Unfortunate Trend in Specialty Coffee Caused by Mislabeling of *Coffea liberica*?

**DOI:** 10.3390/foods10061329

**Published:** 2021-06-09

**Authors:** Dirk W. Lachenmeier, Steffen Schwarz

**Affiliations:** 1Chemisches und Veterinäruntersuchungsamt (CVUA) Karlsruhe, Weissenburger Strasse 3, 76187 Karlsruhe, Germany; 2Coffee Consulate, Hans-Thoma-Strasse 20, 68163 Mannheim, Germany; schwarz@coffee-consulate.com

**Keywords:** coffee, fermentation, gastrointestinal tract, Kopi Luwak, civet

## Abstract

In the context of animal protection, the trend of digested coffees such as Kopi Luwak produced by civet cats in captivity should not be endorsed. Previous studies on such coffees may have been flawed by sample selection and misclassification. As wild civets may prefer *Coffea liberica* beans, due to their higher sugar content, the chemical differences may be caused by the *Coffea* species difference combined with a careful selection of ripe, defect-free cherries by the animals, rather than changes caused by digestion. This may also explain the observed differences between Kopi Luwak from wild civets (mainly *C. liberica*) compared to the one from animals in captivity (typically fed with *C. arabica* and/or *C. canephora*).

## 1. Introduction

The topic of digested coffees is currently receiving a renewed interest and has recently been proposed as a “new trend in specialty coffee” [1].

In this commentary, we want to point out the interesting issue of *Coffea* species assignment in the context of digested coffee studies. The first problem emerges when chemical studies are conducted in non-coffee growing countries, and sampling relies on commercial suppliers, often with doubtful authenticity. The control group is also problematic in the digested coffee studies, as wild civets may select the sweetest, most ripe, and healthy cherries, while the control coffee of commercial quality may include different stages of ripeness and the typical amount of defective beans. For example, it makes no sense to use a Brazilian *C. arabica* coffee as control group for Kopi Luwak from Indonesia. Geographical and variety differences within *C. arabica* alone may explain the observed differences.

The second problem with digested coffees is that many studies may have missed that the actual coffee species under investigation has been *Coffea liberica*, not *Coffea arabica* or *Coffea canephora*, which has been incorrectly assumed. This hypothesis was first raised during an international roasting competition for Liberica coffee [2].

## 2. A Short Critique of Previous Digested Coffee Studies

The study of Marcone [3] is currently the most widely cited study on digested coffees according to Google Scholar (187 citations in June 2021). Marcone [3] obtained Kopi Luwak and control beans (not having gone through the palm civet) from a supplier in California. Both the Kopi Luwak and control coffee beans were claimed as being Indonesian *Coffea canephora* var. robusta. The study also included African civet coffee collected in western Ethiopia. No species was provided for the Ethiopian coffee, which, however, should be assumed as being *Coffea arabica*, the predominant species in Ethiopia. Marcone [3] provided photographs of the studied beans (reproduced in Figure 1a–c).

According to Marcone [3], the beans were assigned as *C. canephora* (Figure 1a), and two types of Ethiopian coffee (Figure 1b,c). However according to our assessment of the shapes, the beans are actually *C. liberica* (Figure 1a), *C. canephora* (which is rather unusual for Ethiopia, therefore assumed as an adulterated product) (Figure 1b), and *C. arabica* (Figure 1c). Please note the bulging and raised nature of the beans at the cut for liberica (Figure 1a). Arabica and canephora are flat at the cut and equally high on both sides. In our opinion, the mislabeling is quite clear. For comparison purposes, we provide examples of authentic *C. liberica* (Figure 1d), *C. canephora* (Figure 1e) and *C. arabica* (Figure 1f). The fact that *C. liberica* exhibits such a little-noticed existence is surely one of the reasons why this circumstance escaped the authors of Kopi Luwak studies and reviews [1] thus far. The species difference may also explain the different surface morphology of the beans [3]. The discrimination ability of some analytical methods can also be explained in that two different coffee species were compared against each other (i.e., *C. liberica* in Kopi Luwak vs. *C. arabica* as control group, e.g., compare Jumhawan et al. [4,5,6] and Suhandy and Yulia [7]).

## 3. Kopi Luwak a *Coffea liberica* in Disguise?

The distinctly different taste and highly valued flavor of Kopi Luwak coffee may be caused by the pure fact that it is *Coffea liberica*, which has a completely different flavor, with very complex profile compared to the commercial coffee species *C. arabica* and *C. canephora*. *C. liberica* has the highest sugar content of all coffees, and thus has the highest risk of fermentation. The sugar content may also be the reason that the civets and other coffee consuming animals prefer *C. liberica* over the other species, if they are available in the same area.

Diligently prepared *C. liberica* shows intense fruity and floral notes (strawberry, jackfruit, mango, banana) and a lactic character (yogurt, cream, mascarpone, crème fraiche) with a pronounced body and intense sweetness. When roasted too dark, the coffee offers notes that reach into the realm of ripe, sweet blue cheese and cheddar.

The lactic, cheesy, perhaps also animalic character of *C. liberica* may be easily misinterpreted as an influence potentially caused by animal digestion or intra-animal fermentation (i.e., the alleged change in taste caused by digestive enzymes of the animals), which are not convincingly proven in previous scientific studies. Currently, there are no sensory or chemical studies available investigating the possibility to distinguish Kopi Luwak from regular coffee prepared from *Coffea liberica* species.

One of the first descriptions of Kopi Luwak, from Brehm in 1883 [8], suggested that the civet released the undigested seeds, that the excrement consisted entirely of caked, but incidentally undamaged coffee beans, and that the animals provide the very best coffee because they ate the ripest fruits. This description stands largely unchallenged to this day, and the scientific proof for the alternative hypothesis, that animal digestion actually changes the coffee and its flavor profile, so far lacks convincing proof. Due to the animal cruelty involved, we believe that this question does not necessarily need further investigation. Ripe and sweet coffee cherries of *C*. *liberica* may be selected by means other than the use of animals.

## 4. Conclusions

The authors believe that digested coffee is rather a perverted trend in specialty coffee, especially if the civet cats are kept in captivity purely for the purpose of coffee production [9] (Figure 2). In this regard, it is almost a relief that much coffee labelled as “Kopi Luwak” is probably a counterfeited product that has never seen the digestive tract of an animal (42% of Kopi Luwak were claimed as being found to be either complete fakes or adulterated with regular coffee beans [10]).

Hopefully, the observation that an already valued specialty coffee such as Kopi Luwak may actually be *Coffea liberica* will encourage a new debate on this species in coffee cultivation, especially against the backdrop of climate change. It would certainly be desirable for the diversity of flavors in coffee, as well as avoid animal cruelty for an unnecessary procedure.

## Figures and Tables

**Figure 1 foods-10-01329-f001:**
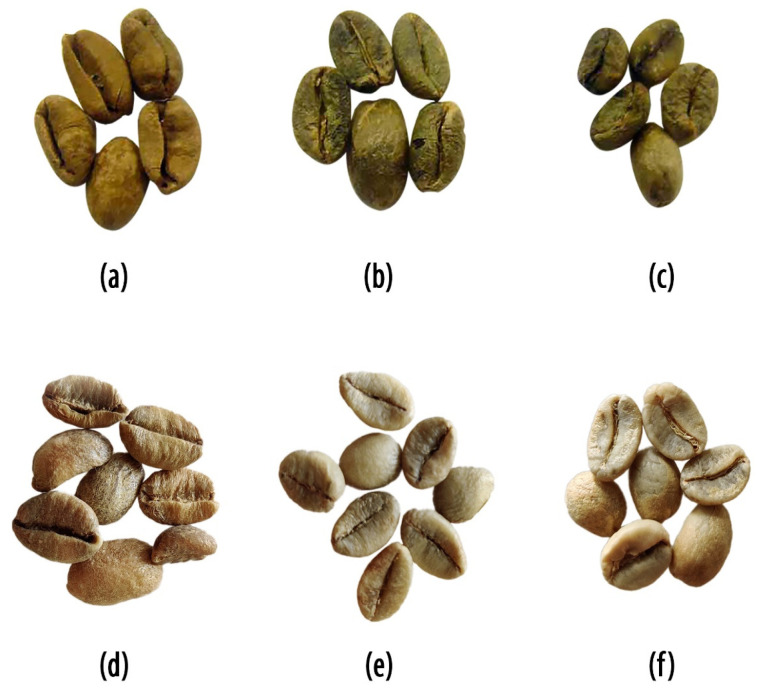
Photographs of coffees claimed as being digested: (**a**) Kopi Luwak coffee beans (claimed as being *Coffea canephora* var. robusta), (**b**) Nekemte-African Civet coffee beans, and (**c**) Abdela-African Civet coffee beans. Photographs of non-digested coffees for comparison: (**d**) *Coffea liberica,* (**e**) *Coffea canephora* var. Old Paradenia (India), and (**f**) *Coffea arabica* var. Catuaí Vermelho (Brasil). ((**a**–**c**) reprinted with graphical improvement (background and noise removed) from Food Research International, 37, Massimo F. Marcone, Composition and properties of Indonesian palm civet coffee (Kopi Luwak) and Ethiopian civet coffee, pp. 901–912 [3], Copyright (2004), with permission from Elsevier. (**d**–**f**) are original photographs).

**Figure 2 foods-10-01329-f002:**
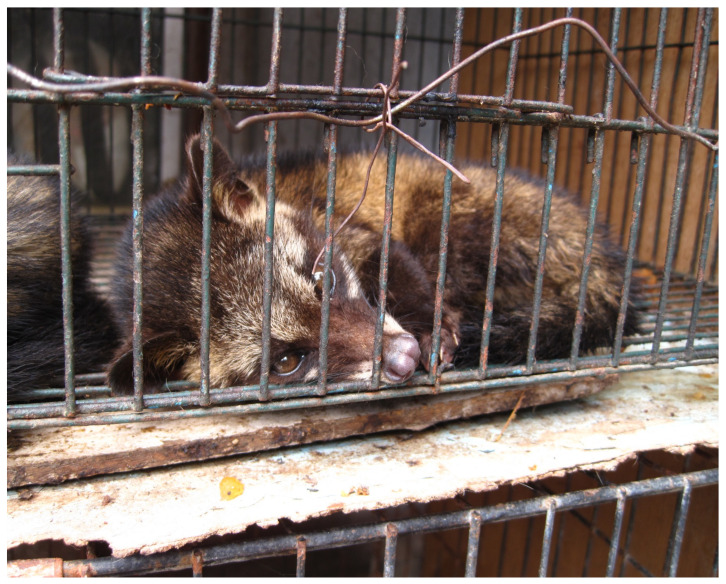
Civet kept caged for Kopi Luwak production (attribution: author Surtr, (https://commons.wikimedia.org/wiki/File:Luwak_(civet_cat)_in_cage.jpg accessed on 8 June 2021) license CC BY-SA 2.0, (https://creativecommons.org/licenses/by-sa/2.0/ accessed on 8 June 2021) via Wikimedia Commons).

## Data Availability

No new data were created or analyzed in this study. Data sharing is not applicable to this article.

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
