# Peer review of "Digested Civet Coffee Beans (Kopi Luwak)—An Unfortunate Trend in Specialty Coffee Caused by Mislabeling of Coffea liberica?"

_foods, 2021, doi:10.3390/foods10061329_

Round 1
Reviewer 1 Report
I have no special remarks to this article/commentary, which points to a justifiable problem of production of Kopi Luwak coffee and the honesty of coffee producers. I agree that the production and labeling of Kopi Luwak requires more control in terms of fairness and authenticity. The high price of such coffee increases the likelihood of fraud and producers may bypass the condition that label must not be confusing to the consumer as to the name of the product and its origin (in the European Union - regulation No 1169/2011 on the provision of food information to consumers)
Perhaps future analytical methods will be able to capture the differences between coffee digested by animals and regular coffee.
Another question that arises is whether, in sensory tests, people are able to distinguish Kopi Luwak from regular coffee prepared from the Coffea liberica species. Are there such studies available in the literature, if not, then it is worth conducting them.
I share the authors' opinion on the protection of animals and perhaps such coffee production should be completely prohibited.
Author Response
I have no special remarks to this article/commentary, which points to a justifiable problem of production of Kopi Luwak coffee and the honesty of coffee producers. I agree that the production and labeling of Kopi Luwak requires more control in terms of fairness and authenticity. The high price of such coffee increases the likelihood of fraud and producers may bypass the condition that label must not be confusing to the consumer as to the name of the product and its origin (in the European Union - regulation No 1169/2011 on the provision of food information to consumers)
Perhaps future analytical methods will be able to capture the differences between coffee digested by animals and regular coffee.
RESPONSE: Thank you for the assessment of our paper.
Another question that arises is whether, in sensory tests, people are able to distinguish Kopi Luwak from regular coffee prepared from the Coffea liberica species. Are there such studies available in the literature, if not, then it is worth conducting them.
RESPONSE: We are not aware about such studies. We have included this point in the text around line 83.
I share the authors' opinion on the protection of animals and perhaps such coffee production should be completely prohibited.
RESPONSE: We completely agree. Probably research as detailed above would not get ethical clearance as well.
Reviewer 2 Report
This is well prepared and important commentary concerning digested coffee labeled as Kopi Luwak. The authors rightly concluded that probably Kopi Luwak is a counterfeited product and does not pass through the digestive tract of a civet cat. Kopi Luwak coffee is probably Coffea liberica. Such observation is very important in the context of coffee cultivation. Importantly, it also allows to avoid civet cat cruelty.
I have only one comment. The number of references should not exceed 10.
Author Response
This is well prepared and important commentary concerning digested coffee labeled as Kopi Luwak. The authors rightly concluded that probably Kopi Luwak is a counterfeited product and does not pass through the digestive tract of a civet cat. Kopi Luwak coffee is probably Coffea liberica. Such observation is very important in the context of coffee cultivation. Importantly, it also allows to avoid civet cat cruelty.
RESPONSE: Thank you for the evaluation of our paper.
I have only one comment. The number of references should not exceed 10.
RESPONSE: One reference was deleted to bring the number to 10. Thank you for pointing our this issue.
Reviewer 3 Report
Dear Authors,
this is a really interesting commentary which present an in-depth analysis of a new (but debatable) trend in coffee beans sector.
I have only a minor request. The Authors commented that Figure 1 from Marcone contains three misleading examples of digested coffee beans. They support their conclusion based on the comparison of three undigested coffee beans of the type C. liberica (panel c), C. canephora (panel d) and C. arabica (panel e). Actually, such comparison would be objective if based on the comparison with digested coffee beans, at least for C. liberica vs C. canephora.
Without a varietal comparison between similarly digested beans, conclusions on the "striking different color" may be simply explained with the effect of the bean' digestion, rather than with phenotype differences. In pother words, the comparison should be based in similar samples (i.e. all digested beans) and not on three varieties of digested samples vs three other possible varieties but undigested.
Except this point that should be better explained, the commentary is overall intriguing.
Author Response
Dear Authors,
this is a really interesting commentary which present an in-depth analysis of a new (but debatable) trend in coffee beans sector.
I have only a minor request. The Authors commented that Figure 1 from Marcone contains three misleading examples of digested coffee beans. They support their conclusion based on the comparison of three undigested coffee beans of the type C. liberica (panel c), C. canephora (panel d) and C. arabica (panel e). Actually, such comparison would be objective if based on the comparison with digested coffee beans, at least for C. liberica vs C. canephora.
RESPONSE: We basically agree, but as we do not have access to authentic material, nor did we want to conduct such research for ethical reasons, we cannot provide these pictures at the moment. However, we do not believe that the morphology or shape of the beans is expected to change by the digestion. So we are of the opinion that the classification according to the bulging and raised nature of the beans at the cut should be possible irrespective of the digestion state of the bean.
Without a varietal comparison between similarly digested beans, conclusions on the "striking different color" may be simply explained with the effect of the bean' digestion, rather than with phenotype differences. In pother words, the comparison should be based in similar samples (i.e. all digested beans) and not on three varieties of digested samples vs three other possible varieties but undigested.
RESPONSE: Currently there is no empirical evidence available if the colour of the bean may be changed by the digestion or not. The authors believe that is might not be changed, as the striking differences of the C. liberica species appear to remain during digestion. As we lack authentic digested material for comparison (see previous remark), we currently decided to delete the mentioning of color as further point for distinction. The shape of the bean alone, which should not be influenced by digestion, is clearly providing compelling evidence that the beans a-c may have been misclassified.
Except this point that should be better explained, the commentary is overall intriguing.
RESPONSE: Thank you for the assessment of our paper and your valuable remarks.